# Nano Selenium—Enriched Probiotics as Functional Food Products against Cadmium Liver Toxicity

**DOI:** 10.3390/ma14092257

**Published:** 2021-04-27

**Authors:** Simona Ioana Vicas, Vasile Laslo, Adrian Vasile Timar, Cornel Balta, Hildegard Herman, Alina Ciceu, Sami Gharbia, Marcel Rosu, Bianca Mladin, Laurentiu Chiana, József Prokisch, Maria Puschita, Eftimie Miutescu, Simona Cavalu, Coralia Cotoraci, Anca Hermenean

**Affiliations:** 1Faculty of Environmental Protection, University of Oradea, 24 Gen. Magheru St., 410048 Oradea, Romania; sim_vicas@yahoo.com (S.I.V.); vasilelaslo@yahoo.com (V.L.); timar.adrian@gmail.com (A.V.T.); 2“Aurel Ardelean” Institute of Life Sciences, Vasile Goldis Western University of Arad, 86 Liviu Rebreanu St., 310414 Arad, Romania; baltacornel@gmail.com (C.B.); hildegard.i.herman@gmail.com (H.H.); alina_ciceu@yahoo.com (A.C.); samithgh2@hotmail.com (S.G.); ramrosu@gmail.com (M.R.); biancaonitamaria@gmail.com (B.M.); 3Doctoral School of Biomedical Science, University of Oradea, 1 University St., 410087 Oradea, Romania; laurentiu.chiana@gmail.com (L.C.); 4Institute of Animal Science, Biotechnology and Nature Conservation, Faculty of Agricultural and Food Sciences and Environmental Management, University of Debrecen, 4032 Debrecen, Hungary; jprokisch@agr.unideb.hu (J.P.); 5Faculty of Medicine, Vasile Goldis Western University of Arad, 86 Liviu Rebreanu St., 310414 Arad, Romania; puschita.maria@uvvg.ro (M.P.); miutescu.eftimie@uvvg.ro (E.M.); ccotoraci@yahoo.com (C.C.); 6Faculty of Medicine and Pharmacy, University of Oradea, 10 Pta 1 Decembrie St., 410073 Oradea, Romania

**Keywords:** selenium nanoparticles, *Lactobacillus casei*, cadmium, antioxidant enzymes, liver, histology, anti-apoptotic, anti-inflammatory

## Abstract

Since cadmium is a toxic metal that can cause serious health problems for humans, it is necessary to find bioremediation solutions to reduce its harmful effects. The main goal of our work was to develop a functional food based on elemental selenium nanoparticles (SeNPs) obtained by green synthesis using *Lactobacillus casei* and to validate their ability to annihilate the hepatic toxic effects induced by cadmium. The characterization of SeNPs was assessed by UV–Vis spectroscopy, FTIR, XRD, DLS and TEM. In order to investigate the dose-dependent protective effects of SeNPs on Cd liver toxicity, mice were assigned to eight experimental groups and fed by gavage, with 5 mg/kg b.w. cadmium, respectively, with co-administration with SeNPs or lacto-SeNPs (LSeNPs) in 3 doses (0.1, 0.2 and 0.4 mg/kg b.w.) for 30 days. The protective effect was demonstrated by the restoration of blood hepatic markers (AST, ALT, GGT and total bilirubin) and antioxidant enzymes, such as catalase (CAT) and glutathione peroxidase (GPx). Moreover, the antioxidant capacity of mice plasma by the FRAP assay, revealed the highest antioxidant capacity for the 0.2 mg/kg LSeNPs group. Histopathological analysis demonstrated the morphological alteration in the group that received only cadmium and was restored after the administration of SeNPs or LSeNPs, while the immunohistochemical analysis of the *bcl* family revealed anti-apoptotic effects; the Q-PCR analysis showed an upregulation of hepatic inflammatory markers for the group exposed to Cd and a decreased value for the groups receiving oral SeNPs/ LSeNPs in a dose-dependent manner. The best protective effects were obtained for LSeNPs. A functional food that includes both probiotic bacteria and elemental SeNPs could be successfully used to annihilate Cd-induced liver toxicity, and to improve both nutritional values and health benefits.

## 1. Introduction

Heavy metals are widely found in our environment and have adverse health effects on the human metabolism. Acute heavy metal intoxications may damage central nervous function, the cardiovascular and gastrointestinal systems, the liver and kidney [1,2]. 

Cadmium is an environmental toxicant that presents higher rates of soil-to-plant compared with other heavy metals, making foodstuffs the major source of cadmium exposure for non-smoking consumers [3,4]. EFSA’s Panel on Contaminants in the Food Chain established for cadmium a tolerable weekly intake of 2.5 µg/kg b.w., a level that ensures a high level of protection for consumers [5]. The greatest dietary impact of cadmium occurs when certain foods are consumed in high quantities, such as grains or grain products (26.9%), vegetables (16.0%) and starchy roots and tubers (potatoes and potato products) (13.2%) [4].

Conventional treatment against heavy metals toxicity is based on chelation therapy using different chemical chelators that can have several adverse effects, such as kidney overload, cardiac arrest, mineral deficiency and anemia [6]. In recent years, interesting candidates for the treatment of heavy metal intoxications, including nanoparticles, probiotics, vitamins (C, E), folate, and essential amino acids have been used [7,8,9].

It has been widely accepted that a functional food provides both nutritional values and health benefits. Even though the concept and “functional food” term were first mentioned in Japan, in 1984 [10], it has since undergone some additional European statements: “Food products can only be considered functional if together with the basic nutritional impact it has beneficial effects on one or more functions of the human organism thus either improving the general and physical conditions or/and decreasing the risk of the evolution of diseases” [11]. Therefore, it can be accepted that functional food science emerged as a fusion between food science, nutrition, medicine, and pharmaceutics.

Selenium (Se) is a contradictory mineral, because at high levels it can become toxic for the organism, while its deficiency also produces several health problems [12]. At the same time, Se is an essential nutrient in human life as it is involved in major biochemical reactions in the body and also in the structure of many enzymes or selenoproteins that play important roles in antioxidant pathways, the endocrine and immune system, reproduction, muscle function, and tumor prevention [12,13].

Selenium is classified as a metalloid and elementally as Se, and it has different allotropic forms including rhombohedral, three deep-red monoclinic forms (α-, β-, and γ-Se), trigonal gray Se, amorphous red Se, and black vitreous Se [12]. Meat, seafood and cereals are the most important food sources of Se.

Se exists in different oxidation states including selenate (Se^+6^), selenite (Se^+4^), selenides (Se^−2^) and elemental selenium (Se^0^) [14]. In food products, Se occurs in combination with proteins: the main Se species includes organic Se such as Se-methyl-selenocysteine, γ-glutamyl-Se-methyl-selenocysteine and selenomethionine. Selenocysteine dominates in the products of animal origin. The main sources of Se are foods enriched in proteins, such as meat and dairy products, fish seafood, milk, and nuts. A low level of Se is found in fruits and vegetables [15].

However, the biological and toxicological effects of Se strongly depend on its chemical form, since it was accepted that its organic form is more favorable in terms of bioavailability [16]. The World Health Organization has established a value of 70 μg/day for the maximum daily intake, considering that doses above 400 μg/day may exert toxic actions [13].

On the other hand, by comparison with organic or inorganic selenium compounds, selenium nanoparticles (SeNPs) display better bioavailability, higher biological activity and lower toxicity, as demonstrated by several studies [17,18,19]. In addition to its antioxidant effect, its use in chemopreventive agents and anticancer drugs is also well documented [20,21], along with evidence of its antimicrobial and antifungal properties [22,23]. SeNPs can be synthesized by chemical [24], physical [25] or biological method, also known as green synthesis [26].

The biogenic synthesis of SeNPs as a green, eco-friendly approach, has attracted attention in recent years due to its low cost and simplicity, demonstrating the accumulation and biotransformation of selenium into both organic (seleno-aminoacid) and elemental forms (Se^0^) by lactic bacteria [27]. Different lactic bacteria strains are able to produce SeNPs with a different size, ranging from 50–100 nm (*Streptococcus thermofilus*) and 100–200 nm (*Lactobacillus* sp.) to 400–500 nm (*Bifidobacter* sp.) [27]. Moreover, nanosized selenium in the range of 100–500 nm may help in the bio-fortification of crops (*Brassica* species), with a higher nutritional impact and health benefits, since it was demonstrated that selenium uptake by plants from the soil is strongly related to its selenium forms: elemental selenium, selenite and selenate, in association with other elements or in organic forms [28].

As the microbial transformations of selenium species with relevance to heavy metal bioremediation has been already documented [29], in this study we obtained, characterized and validated the SeNPs obtained by green synthesis using *Lactobacillus casei*. Moreover, we proposed investigating for the first time the protective effect of SeNPs and lacto-SeNPs (LSeNPs) administered orally to mice for 30 days in different concentrations (0.1, 0.2 and 0.4 mg/kg b.w.), against the toxic effects exerted by cadmium at the hepatic level. Blood biochemical parameters (transaminases, bilirubin, gamma glutamyl transferase), antioxidant enzymes (catalase and glutathione peroxidase), the antioxidant capacity of plasma along with the histology, immunohistochemistry for mitochondrial apoptosis markers (*bcl-2*, *bax*) and gene expression of hepatic inflammatory markers (NF-ĸB, TNFα, IL-6) were analyzed in terms of the comparative evaluation of the dose-dependent protective activity of SeNPs and LSeNPs against cadmium intoxication.

## 2. Materials and Methods

### 2.1. Biosynthesis and Characterization of SeNPs

*Lactobacillus casei* (Lyofast LC4P1, Sacco, Cadorago, Italy) was selected for SeNPs synthesis via a reduction route using sodium hydrogen selenite (NaHSeO_3_) as a reducing agent, according to a protocol described by Eszenyi et al. [27]. In this study, two products were obtained: purified nanoselenium (SeNPs) and lacto-nanoselenium (LSeNPs).

MRS culture medium was inoculated with *L. casei* and sodium hydrogen selenite at a concentration of 200 mg/L in order to promote SeNPs synthesis. Instead, in order to obtain LSeNPs, the culture medium was replaced with skimmed milk. The reaction was allowed to start in a fermentation bottle during 48 h at 37 °C until the characteristic red color of the elemental nano-selenium was achieved. Then, the bacterial cells were removed from the mixture by centrifugation at 6000 rpm, for 15 min, the supernatant was discarded and the pellet was recovered in distilled water. As the mechanism of elemental Se formation is mainly intracellular for lactic acid bacteria [27], the acid digestion was performed in order to remove the bacterial cell wall. After washing, vacuum filtering and the freeze-drying procedure, the collected precipitate was characterized by TEM (transmission electron microscopy, TecnaiG2 F30 S-TWIN, FEI, Frankfurt, Germany), XRD (X-ray diffraction, Mini Flex 600, Rigaku, Tokyo, Japan, operating at 40 kV, 15 mA, with CuK α monochromatic radiation) and FT-IR spectroscopy (Fourier transform infrared spectroscopy, Spectrum BXII spectrophotometer, Perkin Elmer equipped with MIRacle ATR accessory, at scanning speed of 32 cm^−1^ and spectral width 2.0 cm^−1^, Buckinghamshire, UK). For DLS (dynamic light scattering, ZEN 3690, Malvern Instruments, Malvern, Worcestershire, UK) and zeta potential measurements, the SeNPs powders were resuspended in distillated water and sonicated during 10 min before each measurement to prevent aggregation.

### 2.2. Animal and Experimental Design

Six- to eight-week-old CD1 female mice, weighing 26 ± 3 g, were used for the experiments. The mice were housed in a controlled microclimate environment, with a dark–light cycle of 12/12 h, and watering and feeding ad libitum. Mice were fed with an autoclavable standard scientific diet for rodents (Safe D40 diet, SAFE Complete Care Competence, Germany), which is certified as free of toxic substances and balanced regarding the content of amino acids, fatty acids, minerals, and vitamins.

All experimental procedures were approved by the Ethical Committee of the “Vasile Goldis” Western University of Arad and certified by the National Sanitary Veterinary and Food Safety Authority of Romania (005/02.27.2017). We chose females for the animal model, considering the larger study this work is part of which seeks to evaluate multi-organ effects, and knowing that cadmium is more retained in their body compared to males, due to estrogenic effects [30].

The mice were divided into 8 experimental groups (n = 10), as follows: Group 1 (control group), where only the vehicle was administered by gavage (water); Group 2, which received orally CdCl_2_ (Cd group) at a dose of 5 mg/kg b.w.; Group 3 (SeNPs 0.1 group), 4 (SeNPs 0.2 group) and 5 (SeNPs 0.4 group) which were given purified SeNPs in 3 different doses: 0.1, 0.2 and 0.4 mg/kg b.w., respectively, together with 5 mg/kg b.w. of Cd for each group. Groups 6 (LSeNPs 0.1 group), 7 (LSeNPs 0.2 group) and 8 (LSeNPs 0.4 group) received LSeNPs at doses of 0.1, 0.2 and 0.4 mg/kg b.w., respectively, together with 5 mg/kg b.w. of Cd for each group. The administration of SeNPs and LSeNPs was performed one hour after the administration of Cd.

The three doses of SeNPs (0.1, 0.2, 0.4 mg/kg b.w.) and the route of administration were selected according to the results in which protection was obtained against cadmium, administered to mice at a dose of 5 mg/kg b.w. [31].

Thirty days after the first oral administration, the mice were euthanized under anesthesia with a mixture of ketamine and xylazine. Blood and liver tissues were collected for further analysis.

### 2.3. Blood Biochemical Parameters

Blood samples were collected by cardiac puncture. The samples were centrifuged at 3500 rpm for 10 min. Samples were analyzed for aspartate aminotransferase (AST), alanine aminotransferase (ALT), gamma-glutamyltransferase (GGT) and total bilirubin, levels (ChemaDiagnostica, Monsano, Italy) with a Mindray BS-120 Chemistry Analyzer (ShenzenMindray Bio-Medical Electronics Co., Ltd., Nanshan, Shenzhen, China).

### 2.4. Antioxidant Enzymes Assay

Antioxidant enzyme catalase (CAT) was determined from liver samples using a commercially available Catalase Activity Colorimetric Assay kit (Canvax Biotech, S.L., Córdoba, Spain). The absorbance of samples was measured at 570 nm, using a Tecan microplate reader. The results were expressed as nmol of H_2_O_2_ decomposed by catalase in 30 min reactions.

Glutathione peroxidase (GPx) was determined from liver samples using the glutathione peroxidase activity kit (Enzo, Catalog No. ADI-900-158) and the rate of decrease in the absorbance at 340 nm was directly proportional with glutathione peroxidase activity in the samples. The GPx activity was expressed as Units/mL.

### 2.5. Antioxidant Capacity of Mice Plasma—FRAP Assay

The FRAP values of mice plasma were determined according to Benzie and Strain, 1996 [32], with some modification. All reagents were prepared and used the same day. The working FRAP reagent was prepared by mixing in ratio 10:1:1 (*v/v/v*) of 300 mM acetate buffer (pH 3.6), 10 mM TPTZ (2,4,6-tripyridyl-s-triazine) in 40 mM HCl and 20 mM FeCl_3_·6H_2_O. Briefly, a 50 µL sample was mixed with 150 µL distilled water and 1.5 mL working FRAP solution. After 30 min, the absorbance at 595 nm was measured against a reagent blank at 37 °C using The PharmaSpec Shimadzu UV–Vis 1700 (Shimadzu, Kyoto, Japan) spectrophotometer. The FRAP values were expressed as mmol FeSO_4_/L by using different concentrations of aqueous solutions of FeSO_4_·7H_2_O (in the range of 50–1000 µM) for a standard curve.

### 2.6. Histopathology Analysis

Formalin-fixed samples of the liver were dehydrated with gradient solutions of ethanol, embedded in paraffin and then sectioned at 5 µm. Slides were stained with hematoxylin and eosin (H&E) for routine histological evaluation. Mounted sample slides were examined under an Olympus BX43 light microscope (Tokyo, Japan) and the images were captured using an XC30 CCD camera (Tokyo, Japan).

Histopathological changes were qualitatively described, and the alterations were graded (n = 10), using a modified scale from our previous studies [33]:

Grade 1: normal aspect of the hepatocytes and sinusoids;

Grade 2: regularly/irregularly shaped hepatocytes with slightly dilated blood capillaries or blood congestion;

Grade 3: vacuolated hepatocytes, mild dilatation of sinusoids and blood congestion, inflammatory infiltrates;

Grade 4: <5% necrosis, lysis or strong inflammatory changes, large dilatations of sinusoids, increased macrophage number;

Grade 5: >5% necrosis, lysis or strong inflammatory reactions.

### 2.7. Immunohistochemical Analysis

Immunohistochemistry analysis was performed on paraffin embedded 5 μm-thick liver sections. Liver sections were deparaffinized in Dewax (Biosystems, Nussloch, Germany) and rehydrated prior to epitope retrieval in Novocastra sol. (Leica Biosystems, Nussloch, Germany). Following the neutralization of endogenous peroxidase (3% H_2_O_2_), the sections were incubated at 4 °C overnight with anti *bax* and *bcl*-2 antibodies (1:100). Detection was then performed using a polymer detection system (cat. no. RE7280 K; Novolink Max Polymer Detection system) and 3,3’diaminobenzidine (DAB) as a chromogenic substrate. Nuclei were stained with hematoxylin, dehydrated in a gradient of alcohol, and mounted onto slides. The slides were examined and images were captured as described previously.

### 2.8. RT-PCR Analysis

Tissue samples collected for the analysis of the expression of NF-ĸB p65,TNF-α, IL-6 gene involved in the inflammatory process were stored in an RNA Shield solution at a temperature of −80 °C until processing. The total RNA was extracted using an SV Total RNA Isolation System extraction kit, purchased from Promega, according to the manufacturer’s recommendations. The quantitative and qualitative of purified RNA evaluation was assessed spectrophotometrically, using the NanoDrop 8000 spectrophotometer produced by Thermo Scientific, Waltham, MA, USA. The conversion of the total RNA to complementary DNA was performed using 2 micrograms of total RNA and the First Strand cDNA Synthesis Kit conversion kit. To determine the quantitative expression, we used a LuminarisHiGreenqPCT Master Mix kit (Thermo Scientific, Waltham, MA, USA), low ROX, each sample being determined in triplicate. The PCR system used was Applied Biosystems 7500 Real Time PCR System (Foster City, CA, USA). Primers used were included in Table 1. The results obtained were interpreted using the 2ΔΔCT method, proposed by Livak in 2001 [34].

### 2.9. Statistical Analysis

Data were statistically processed using GraphPad Prism 3.03 software (GraphPad Software, Inc., La Jolla, CA, USA), and one-way analysis of variance, followed by a Bonferroni test. *p* < 0.05 was considered to indicate a statistically significant difference.

The data expressed as mean ± standard deviation (SD) and statistically significant differences (* *p* < 0.05 and # *p* < 0.05) were determined compared with control (group 1) and the cadmium group (group 2), respectively.

## 3. Results and Discussion

### 3.1. Physico–Chemical Characterization of SeNPs

The morphology of SeNPS was evidenced by TEM as presented in Figure 1, showing homogenous and spherical shape particles, with a diameter of a maximum 80 nm, as confirmed by DLS. Zeta potential measurement indicated −22 mV with a polydispersity index less than 0.2, indicating a good stability.

A preliminary confirmation of SeNPs formation was achieved by plasmon resonance in UV–Vis spectroscopy, observing the maximum adsorption at 270 nm (Figure 2a). This result is in agreement with some previous papers [35,36]. The SeNPs’ production using different lactic acid bacteria show unique structured nanospheres with regular and uniform size, comparative with chemical synthesis [27,37,38]. To date, different microorganisms (*Lactobacillus* sp. *Bifidobacter* sp. *Streptococcus thermophilus*) are able to reduce inorganic selenium (selenite and selenite oxoanions) into SeNPs in different size, ranging from 50 to 500 nm [24]. The conversion of selenite to SeNPs by microorganism involves different mechanisms, where the reductase enzymes play an important role [29]. The biosynthesis of SeNPs by microorganisms is a cheaper and faster process, where high purity selenium spheres should be produced. Furthermore, the probiotic bacteria and SeNPs are safe for clinical administration and human consumption [24].

In order to determine the functional groups that exist on the surface of SeNPs, FTIR vibrational features were obtained and presented in Figure 2b. The main vibrational bands in the high wavenumber region are 3300 cm^−1^, 2920 cm^−1^ and 2850 cm^−1^ corresponding to the stretching vibration of OH groups, aliphatic C–H and carboxylic acid O–H groups, respectively. In the low wavenumber region, 1720 cm^−1^ is assigned to carbonyl C=O stretch, 1658 cm^−1^ to amide I vibration, 1540 cm^−1^ to amide II and 1230 cm^−1^ to amide III vibrations. The most intense vibrational band at 1040 cm^−1^ represents the characteristic of the Se–O bond stretching, according to previously reported data in the literature [36,39]. These results indicated the presence of both proteins and carbohydrates on the surface of nanoparticles, deriving from the cell membrane, which may improve the long-term stability and protection of the core-particle. The XRD pattern of the Se nanoparticles (Figure 2c) shows a broad peak at about 2θ = 22° suggesting the amorphous structure of SeNPs, in agreement with previous data [24,35,36], which demonstrated that stable amorphous forms (or even low crystallinity) are advantageous for biological applications, as they exhibit better solubility and subsequent adsorption and bioavailability.

The bioconversion of the inorganic Se^+4^ into elemental selenium form by *Lactobacilus casei* was performed in this study to decrease its toxicity and to provide highly bioavailable forms, which can be further used in functional foods. There are some studies which provide data regarding the conversion of inorganic to organic selenium, such as on cabbage and yeast which have been applied for the industrial production of selenium–methionine from inorganic selenium [40]. Moreover, a previous study demonstrated that some anaerobic or semi-anaerobic bacteria can convert the Se^+4^ into elemental selenium ranging between 100 and 500 nm [27,41]. The biological activity of SeNPs depends on their size, meaning that the smaller particles have a higher activity. The SeNPs is selenium in the zero-oxidation state that presents a lower toxicity and very good bioavailability compared to other oxidation states (Se^+4^, Se^+6^) [42].

### 3.2. Effect of SeNPs on Blood Biochemical Parameters

The results of biochemical parameters are shown in Table 2. Blood alanine transaminase (ALT) and aspartate transaminase (AST) activity increased after Cd administration in mice (group 2) by 37.92 and 33.40%, respectively. The treatment with SeNPs or LSeNPs significantly decreased the activity of these enzymes compared to group 2, but not for group 3, where the lowest dose of SeNPs (0.1 mg/kg b.w.) was used. The GGT activity was significantly decreased only for group 3. The total bilirubin level was not affected by the treatment of both forms of SeNPs.

Plasma transaminases are known to be important indicators for assessing the health of liver tissue. Cd leads to an increase in blood transaminase enzymes by transferring them to the blood and destroying membrane permeability due to lipid oxidation [43,44].

As expected, elevated levels of hepatic enzymes (AST and ALT) were recorded after Cd administration compared to the control, which is in line with other similar studies [45,46,47]. Co-administration of Cd and both the form of SeNPs showed that liver markers decreased significantly in a dose-dependent manner compared to the metal-group and highlights the protective effect of both forms of SeNPs against the toxic injuries induced by the metal. The best effects were obtained with LSeNPs.

### 3.3. Effect of SeNPs on Antioxidant Enzyme Assay

Cadmium is absorbed from the gastrointestinal tract and is mainly accumulated in liver and kidney where it is bounded to metallothionein (MT) and provides de novo protection against Cd. When it exceeds the binding capability of MT due to a higher exposure to metal, the non-bounded Cd ions cause hepato- and nephrotoxicity [46]. Moreover, the fact that cadmium stimulates free radical production, resulting in oxidative changes of lipids, proteins and DNA, may in turn suppress hepatic and renal functions and initiate different pathologies [47].

Catalase and glutathione peroxidase are included in the enzymatic antioxidant defense system that protects cells against reactive oxygen species toxicity and lipid peroxidation. Catalase cleaves hydrogen peroxide into water and oxygen [48]. In the case of Cd intoxication, the activity of this enzyme is decreased [43]. Moreover, glutathione peroxidase (GPx) catalyzes the reduction of hydrogen peroxide (or a variety of organic hydroperoxides) with reduced gluthatione to form gluthathione disulfide (GSSG). The low GPx activity is one of the consequences of oxidative stress [49].

Our results showed that Cd alone induced a significant reduction in catalase activity (Figure 3A) compared to the control. After the combination of Cd and SeNPs (groups 3–5), an improvement in catalase activity was recorded compared with cadmium group in a dose-dependent manner. Instead, using LSeNPs at a high dose (4 mg/kg b.w.) results a significantly increased in catalase activity comparative with both the control and cadmium groups. This result demonstrates the beneficial effect of SeNPs in combination with probiotics against Cd toxicity. On the other hand, the treatment with Cd (group 2) showed a significant decrease in GPx activity (*p* < 0.001) compared with the control group (Figure 3B). The treatments using a combination of Cd and SeNPs or LSeNPs at different concentrations also gave a significant decrease in enzyme activity.

The significant decrease in the antioxidant enzymes after exposure to Cd was similar to that of El-Boshy et al. [48] in rats for catalase and GPx, while the sodium selenite-treated group showed significantly increased CAT and GPx activities. The combination of sodium selenite with cadmium enhanced the antioxidant activities of CAT and GPx and ameliorated the cadmium-induced liver damage by improving hepatic markers [48].

The effects of cadmium and sodium selenite on antioxidant enzymes in the liver, kidneys and testes of rats were investigated by Dzobo et al., 2013 [50]. Their results showed that in the liver, Cd treatment resulted in decreased catalase activity, while the Se treatment resulted in increased catalase activity. Instead, the co-treatment of Cd and Se resulted in an increase in GPx.

The major damage caused by toxic metals is due to the production of free radicals, which induce oxidative stress in cells, and cause damages. Oxidative stress is produced when the balance between the antioxidant system and ROS is in favor of free radicals [43].

The antioxidant capacity of selenium could be attributed to its presence in GPx or thioredoxin reductase [42,51]. The protective effect of selenium against cadmium-induced tissue damage could be attributed to its antioxidant activity through the enhancement of antioxidant enzymes from tissues.

To our knowledge, this study was one of the first which demonstrated the hepatoprotective activity of elemental selenium in the form of nanoparticles against toxicity induced by cadmium due to enhanced antioxidant enzyme activities. Several mechanisms may be involved in the protection of selenium. One hypothesis is to change the absorption of cadmium and its distribution in the body and target organs. Another hypothesis takes into account that selenium (in inorganic or organic form), is well known for its ability to eliminate ROS and improve the antioxidant system damaged caused by Cd [48].

### 3.4. Antioxidant Capacity of Mice Plasma

FRAP values are shown in Figure 4. The treatments with Cd and combined Cd+SeNPs did not significantly differ between groups vs. control group. The highest FRAP value was recorded for group 7 that received combination LSeNPs and Cd. This result highlights the synergistic effect between SeNPs and probiotic bacteria regarding the protective effect against Cd toxicity.

In one study [52], red elemental SeNPs (within a size range of 80–220 nm) were biosynthesized using the marine strain of *Bacillus* sp. and in vitro investigated in terms of antioxidant activity by DPPH and the reduction power assay. SeNPs revealed moderate antioxidant activity compared to SeO_2_ and BHT due to the fact that elemental selenium is insoluble.

SeNPs have high antioxidant activity compared with other chemical forms of Se. Wang et al. [42] showed the antioxidant activity of SeNPs that demonstrated lower toxicity compared with selenomethionine.

### 3.5. Histopathology Analysis

The liver is an important organ for metabolism, detoxification, storage, and the excretion of xenobiotics or metabolites, which is vulnerable to injury. As the liver is an important target of cadmium [53], we assessed the structural changes by microscopic analysis.

Light microscopic examination showed a normal structure of the liver (Figure 5, Group 1) in the controls. Exposure to Cd induced degenerative changes in the liver, mainly consisting of focal hepatocyte necrosis and lysis, pycnotic nuclei with condensed chromatin, as well as sinusoidal congestion and parenchyma infiltration by mononuclear cells (Figure 5, Group 2). The co-administration of SeNPs practically prevented the changes in the liver structure in a dose-dependent manner. The best protection was obtained with LSeNPs, where we noticed the presence of rare inflammatory cells in the sinusoids (Figure 5, Groups 6–8). Moreover, histomorphometric evaluation demonstrated that the dose of 0.2 mg/kg and 0.4 mg/kg induced significantly fewer liver structural changes compared to the liver intoxicated with cadmium alone (*p* < 0.001; Figure 6).

Regarding the antagonistic effect of SeNPs and LSeNPs, our results suggest that Se in both forms was able to reduce the hepatotoxicity of cadmium in a dose-dependent manner (Figure 5 and Figure 6). The present study demonstrated that co-treatment with SeNPs ameliorated the histopathological damage induced by Cd in the tissues of liver by reducing the toxicity comparative to control group [54].

### 3.6. The SeNPs Prevent Apoptosis in Liver Parenchyma Induced by Cadmium

Exposure to Cd causes the activation of multiple death signals in parallel, including the activation of apoptosis-related mitochondrial signaling and DNA damage response [55]. Therefore, the members of the *bcl-2* protein family known to regulate the release of apoptosis-activating factors by changing ratio of *bcl-2* to *bax* which determines cell survival or cell death [56], are valuable tools to assess the hepatic protective effects of SeNPs and LSeNPs against the pro-apoptotic activity of Cd.

The immunohistochemical analysis of the pro-apoptotic *bax* and anti-apoptotic *bcl*-2 markers revealed a marked expression of *bax* for the liver (Figure 7) exposed to Cd and a dose-dependent decrease in the groups receiving oral SeNPs and LSeNPs. The reduction in *bax* expression is more significant for LSeNPs. In contrast, the immunopositivity for *bcl-2* was significantly reduced in the livers of the Cd group and restored for SeNPs groups in dose-dependent manner, especially for LSeNPs.

### 3.7. The SeNPs Prevent Inflammation in Liver Parenchyma Induced by Cadmium

To investigate the anti-inflammatory ability of SeNPs and LSeNPs, we measured the mRNA expression for the NF-ĸB, a key transcriptional regulator of the inflammatory response [57], and the major upregulated proinflammatory cytokines in the liver.

The level of gene expression of liver TNF-α, IL-6, NF-ĸB p65 was significantly increased compared to the control in the group in which Cd was administered (Figure 8), as previously described [58,59]. The co-administration of Cd with SeNPs or LSeNPs resulted in decreased gene expression, which was directly correlated with the concentration of both SeNPs forms.

## 4. Conclusions

This study proposed a new functional food based on SeNPs obtained by green synthesis using *Lactobacillus casei,* and investigated for the first time its ability to annihilate hepatic toxicity induced by cadmium, as a solution for liver injury bioremediation. Two forms of elemental SeNPs, purified SeNPs and lactic acid bacteria (*L. casei*) together with endogenous SeNPs (called LSeNPs), were tested in Cd-induced liver toxicity to mice. Co-administration of Cd and both forms of SeNPs showed that the blood transaminases decreased significantly in a dose-dependent manner. In addition, LSeNPs at the highest dose (0.4 mg/kg b.w.) significantly increased the catalase activity comparative to the control and cadmium group. The histopathological damage induced by Cd in the mouse liver was ameliorated upon the co-treatment with both forms of SeNPs. Immunohistochemical analysis revealed a reduction in pro-apoptotic *bax* and an increase in anti-apoptotic *bcl-2* expression, especially for LSeNPs. Additionally, the co-administration of Cd with both forms of SeNPs significantly decreased the gene expression of liver inflammatory markers, with the best effects for LSeNPs. Overall, the best hepatoprotective effects were obtained for LSeNPs. A follow-up study will highlight the gender differences in the hepatoprotective effects of SeNPs on Cd-induced liver toxicity to mice, knowing that males are more susceptible to cadmium-induced hepatotoxicity than females [60,61].

A functional food that includes both probiotic bacteria and elemental SeNPs could be successfully used to annihilate Cd-induced liver toxicity, and to improve both nutritional values and health benefits. In this way, a possible new technology is provided for the food industry, the production of yogurt enriched with selenium nanoparticles produced by lactic acid bacteria with protective effects against heavy metals.

## Figures and Tables

**Figure 1 materials-14-02257-f001:**
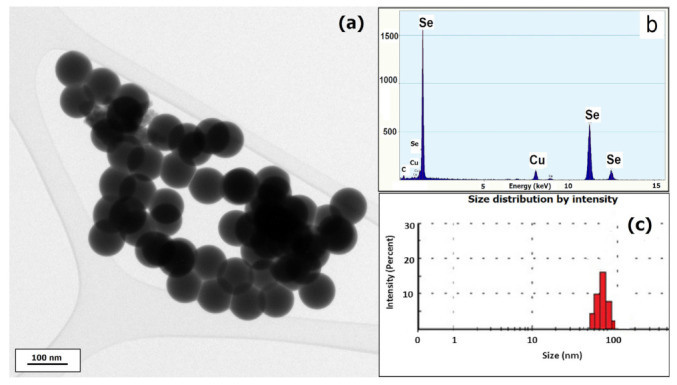
(**a**) TEM micrograph of SeNPs synthesized using *L. casei* and NaHSeO_3_ as a reducing agent; (**b**) energy dispersive X-ray analysis; and (**c**) particles size distribution confirmed by DLS.

**Figure 2 materials-14-02257-f002:**
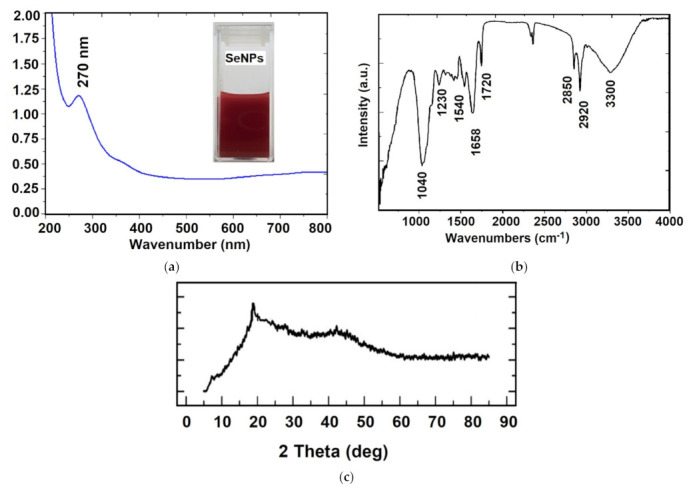
Physico–chemical characterization of SeNPs by UV–Vis spectroscopy (**a**); FTIR spectroscopy (**b**); and XRD pattern (**c**).

**Figure 3 materials-14-02257-f003:**
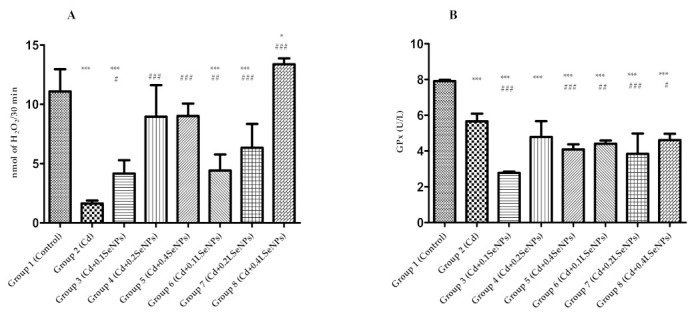
Effects of cadmium and selenium on catalase (**A**) and GPx (**B**) after 30 days of treatment. All values were expressed as the mean ± SD for 10 mice in each group. SeNPs+Cd and LSeNPs+Cd at different concentrations vs. control group: * *p* < 0.05; *** *p* < 0.001. SeNPs+Cd and LSeNPs+Cd at a different concentration vs. the cadmium group: # *p* < 0.05; ## *p* < 0.01; ### *p* < 0.001. Groups: 1—control; 2—CdCl_2_; 3—CdCl_2_ + 0.1 mg/kg SeNPs; 4—CdCl_2_ + 0.2 mg/kg SeNPs; 5—CdCl_2_ + 0.4 mg/kg SeNPs; 6—CdCl_2_ + 0.1 mg/kg LSeNPs; 7—CdCl_2_ + 0.2 mg/kg LSeNPs; 8—CdCl_2_ + 0.4 mg/kg LSeNPs.

**Figure 4 materials-14-02257-f004:**
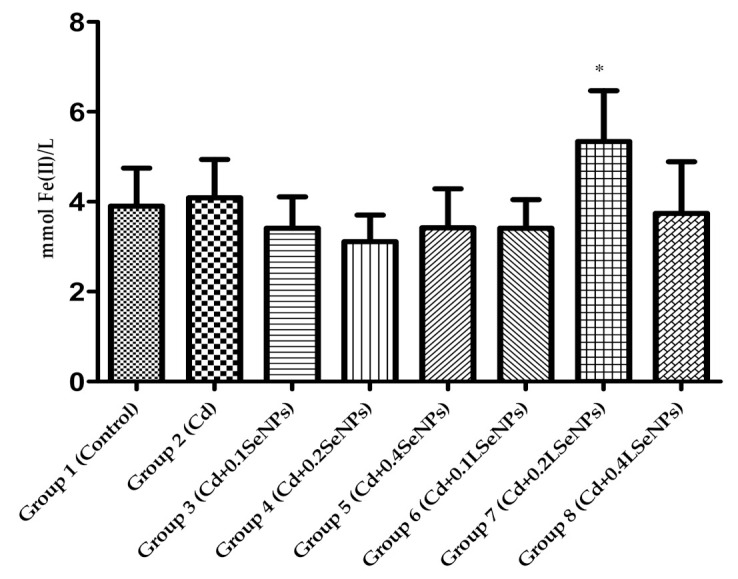
FRAP values of blood mice after 30 days treatments. All values were expressed as mean ± SD for 10 mice in each group. * *p* < 0.05 is statistically significant difference of sample vs. control group. Groups: 1—control; 2—CdCl_2_; 3—CdCl_2_ + 0.1 mg/kg SeNPs; 4—CdCl_2_ + 0.2 mg/kg SeNPs; 5—CdCl_2_ + 0.4 mg/kg SeNPs; 6—CdCl_2_ + 0.1 mg/kg LSeNPs; 7—CdCl_2_ + 0.2 mg/kg LSeNPs; 8—CdCl_2_ + 0.4 mg/kg LSeNPs.

**Figure 5 materials-14-02257-f005:**
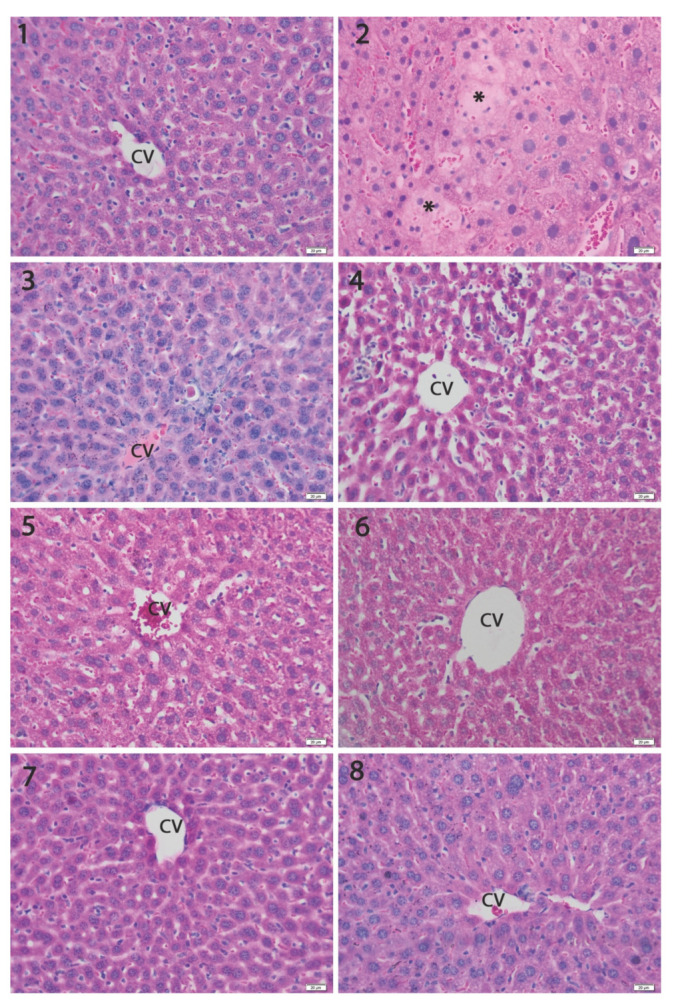
The histopathological sections of mice livers of the experimental groups, H&E barr 20 µm. Various degrees of histopathological changes were observed in Cd exposure groups, mainly including focal hepatocyte necrosis/lysis (*) and degeneration. The co-administration of SeNPs with Cd reduced the metal toxic histological changes in the liver in a dose-dependent manner, which was more obvious for LSeNPs. CV-centrilobular vein. Groups: 1—control; 2—CdCl_2_; 3—CdCl_2_ + 0.1 mg/kg SeNPs; 4—CdCl_2_ + 0.2 mg/kg SeNPs; 5—CdCl_2_ + 0.4 mg/kg SeNPs; 6—CdCl_2_ + 0.1 mg/kg LSeNPs; 7—CdCl_2_ + 0.2 mg/kg LSeNPs; 8—CdCl_2_ + 0.4 mg/kg LSeNPs.

**Figure 6 materials-14-02257-f006:**
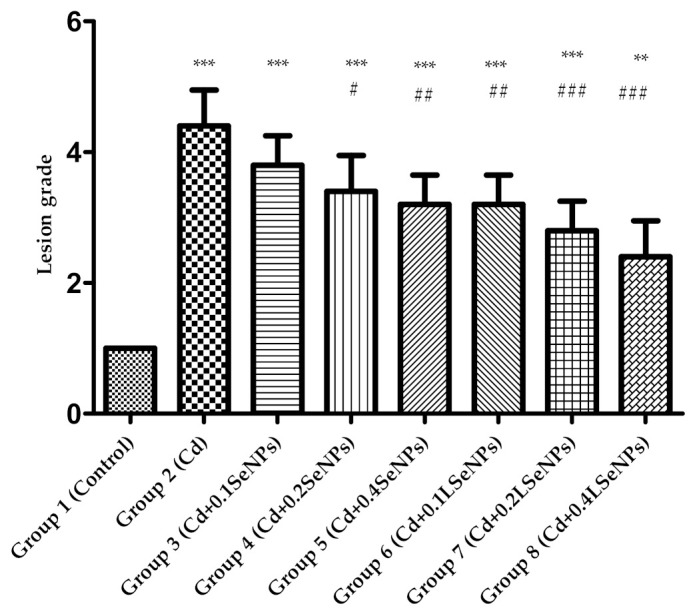
Semi-quantitative assessment of the severity of liver damage ranked from 1 (control status) to 5 (liver damage) induced by cadmium under the protection of SeNPs and LSeNPs. Groups: 1—control; 2—CdCl_2_; 3—CdCl_2_ + 0.1 mg/kg SeNPs; 4—CdCl_2_ + 0.2 mg/kg SeNPs; 5—CdCl_2_ + 0.4 mg/kg SeNPs; 6—CdCl_2_ + 0.1 mg/kg LSeNPs; 7—CdCl_2_ + 0.2 mg/kg LSeNPs; 8—CdCl_2_ + 0.4 mg/kg LSeNPs. ** *p* < 0.01; *** *p* < 0.001 compared to control group (group 1); # *p* < 0.05; ## *p* < 0.01; ### *p* < 0.001 compared to Cd group (group2).

**Figure 7 materials-14-02257-f007:**
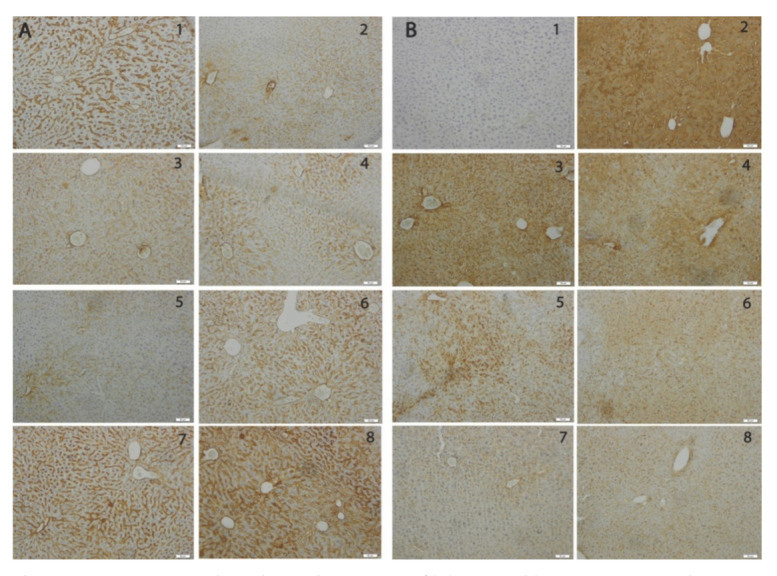
Hepatic immunohistochemical expression of *bcl-2* (**A**) and *bax* (**B**) in: 1—control; 2—CdCl_2_; 3—CdCl_2_ + 0.1 mg/kg SeNPs; 4—CdCl_2_ + 0.2 mg/kg SeNPs; 5—CdCl_2_ + 0.4 mg/kg SeNPs; 6—CdCl_2_ + 0.1 mg/kg LSeNPs; 7—CdCl_2_ + 0.2 mg/kg LSeNPs; 8—CdCl_2_ + 0.4 mg/kg LSeNPs.

**Figure 8 materials-14-02257-f008:**
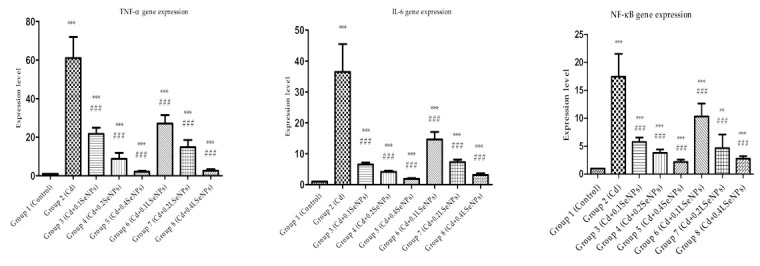
Hepatic TNF-α, IL-6, NF-ĸB p65 gene expressions. *** *p* < 0.001, ** *p* < 0.01 compared to control, ### *p* < 0.001 compared to Cd group. Groups: 1—control; 2—CdCl_2_; 3—CdCl_2_ + 0.1 mg/kg SeNPs; 4—CdCl_2_ + 0.2 mg/kg SeNPs; 5—CdCl_2_ + 0.4 mg/kg SeNPs; 6—CdCl_2_ + 0.1 mg/kg LSeNPs; 7—CdCl_2_ + 0.2 mg/kg LSeNPs; 8—CdCl_2_ + 0.4 mg/kg LSeNPs.

**Table 1 materials-14-02257-t001:** Primer sequences for RT-PCR.

Target	Sense	Antisense
NF-ĸB 65	5′CTTGGCAACAGCACAGACC3′	5′GAGAAGTCCATGTCCGCAAT3′
TNF-α	5′CTGTAGCCCACGTCGTAGC3′	5′TTGAGATCCATGCCGTTG3′
IL-6	5′AAAGAGTTGTGCAATGGCAATTCT3′	5′AAGTGCATCATCGTTGTTCATACA3′
GAPDH	5′CGACTTCAACAGCAACTCCCACTCTTCC3′	5′TGGGTGGTCCAGGGTTTCTTACTCCTT3′

**Table 2 materials-14-02257-t002:** Effects of cadmium and selenium on biochemical parameters after 30 days treatments ^1^.

Biochemical Parameters	Group 1 (Control)	Group 2 (Cd)	Group 3 (Cd + 0.1SeNPs)	Group 4 (Cd + 0.2SeNPs)	Group 5 (Cd + 0.4SeNPs)	Group 6 (Cd + 0.1LSeNPs)	Group 7 (Cd + 0.2LSeNPs)	Group 8 (Cd + 0.4LSeNPs)
AST (U/L)	106.91 ± 16.50	142.62 ± 42.39	131.19 ± 42.44	74.71 ± 4.36 ^###^	74.20 ± 12.19 ^###^	79.09 ± 10.71 ^###^	88.66 ± 30.46 ^##^	78.60 ± 16.61 ^###^
ALT (U/L)	57.15 ± 5.89	78.82 ± 14.17	72.32 ± 17.23	38.62 ± 8.42 ^###^	35.74 ± 11.97 ^###^	34.16 ± 7.73 *^, ###^	35.53 ± 10.91 *^, ###^	33.43 ± 7.16 *^, ###^
GGT (U/L)	2.37 ± 0.31	2.62 ± 0.54	1.35 ± 0.71 *^,##^	2.5 ± 0.10	2.35 ± 0.42	2.02 ± 0.63	2.40 ± 0.73	2.96 ± 0.23
Total bilirubin (mg/dL)	0.19 ± 0.11	0.15 ± 0.04	0.22 ± 0.07	0.22 ± 0.06	0.19 ± 0.07	0.16 ± 0.05	0.15 ± 0.04	0.099 ± 0.07

^1^ All values were expressed as the mean ± SD for 10 mice in each group. Groups: Cd + SeNPs and Cd + LSeNPs at different concentration vs. control group: * *p* < 0.05. Groups: Cd + SeNPs and Cd + LSeNPs at different concentration vs. cadmium group: ## *p* < 0.01; ### *p* < 0.001. Groups: 1—control; 2—CdCl_2_; 3—CdCl_2_ + 0.1 mg/kg SeNPs; 4—CdCl_2_ + 0.2 mg/kg SeNPs; 5—CdCl_2_ + 0.4 mg/kg SeNPs; 6—CdCl_2_ + 0.1 mg/kg LSeNPs; 7—CdCl_2_ + 0.2 mg/kg LSeNPs; 8—CdCl_2_ + 0.4 mg/kg LSeNPs. In the groups 2 to 8, orally CdCl_2_ at a dose of 5 mg/kg b.w. was administrated.

## Data Availability

The data presented in this study are available on request from the corresponding author. The data are not publicly available due to privacy.

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
