# Peer review of "Nano Selenium—Enriched Probiotics as Functional Food Products against Cadmium Liver Toxicity"

_materials, 2021, doi:10.3390/ma14092257_

Round 1

Reviewer 1 Report

ABSTRACT

  • the abstract should start with some general statements and “introduction”.
  • it this clearly cadmium-induced toxicity?
  • give clearly the doses used in this study.
  • clearly give the novelty of this study and highlight significant changes.

INTRODUCTION

  • generally, well prepared with some suggestions:
  • some other aspect are missing – the papers publisher in Toxics should be used: https://www.mdpi.com/journal/toxics/special_issues/metals_diseases

https://www.mdpi.com/journal/toxics/special_issues/Toxicity-Cadmium

  • give also data referred by EFSA
  • why authors focus only on liver structure? Also, some other organs (kidney, repro organs etc.) are interesting site of Cd MoA

MATERIAL AND METHODS

  • well prepare and clear
  • give exactly the blood sampling site

RESULTS AND DISCUSSION

  • the statement (L204-206) should be a hypothesis and this should be transferred to the aim (hypothesis) of study in the final part of introduction
  • for such a study also some morphometric data (histological quantification - as relative volume of structures, diameter of CV, number of cells etc.) should support the effects

CONCLUSION

  • clear, but give also the novelty of this study

SUGGESTION

  • the MS is interesting with some significant results, but there are some imperfection that authors should correct before possible progress

OTHER COMMENTS

  • it is not clear if the MS is correctly related to aim of the journal (this should be decided by editor), but the MS is interesting describing some new ideas

Author Response

Response to Reviewer 1

The authors would like to thanks for Reviewer 1 comments that improve our manuscript quality. 

 (x) English language and style are fine/minor spell check required

The manuscript was checked for type-errors and English reviewed.

Comments and Suggestions for Authors

ABSTRACT

  • the abstract should start with some general statements and “introduction”.

In the abstract was introduced an introductory phrase.

  • it this clearly cadmium-induced toxicity?

We rephrase the abstract and introduced “Cd liver toxicity” and in the materials and methods section we support the Cd dose used and the selection of the SeNPs doses and route of administration.

  • give clearly the doses used in this study.

We introduce in abstract the doses of SeNPs and LSeNPs used in this study.

  • clearly give the novelty of this study and highlight significant changes.

The novelty of the study was highlighted into the introduction and conclusions.

INTRODUCTION

  • some other aspect are missing – the papers publisher in Toxics should be used: https://www.mdpi.com/journal/toxics/special_issues/metals_diseases

https://www.mdpi.com/journal/toxics/special_issues/Toxicity-Cadmium

give also data referred by EFSA

The recommended papers include just data referring to kidney toxicity and other organs, but not to hepatic tissue. However, we introduced some data of EFSA ( EFSA Journal 2011, 2012) and one review from special issues of Toxics (Satarug, 2018), which addressed specifically to liver toxicity.

  • why authors focus only on liver structure? Also, some other organs (kidney, repro organs etc.) are interesting site of Cd MoA

 The investigation of ability nanoparticles and probiotics to annihilate the Cd toxic effects on other organs will be performed in further studies.

MATERIAL AND METHODS

well prepare and clear

  • give exactly the blood sampling site

Blood samples were collected by cardiac puncture.

RESULTS AND DISCUSSION

  • the statement (L204-206) should be a hypothesis and this should be transferred to the aim (hypothesis) of study in the final part of introduction

We changed the statement as directed.

  • for such a study also some morphometric data (histological quantification - as relative volume of structures, diameter of CV, number of cells etc.) should support the effects

We made the semi-quantitative assessment of the severity of liver damage by ranking from 1 (control status) to 5 (liver damage) induced by cadmium under protection of SeNPs and LSeNPs, using a modified scale from our previous studies (Hermenean et al, 2015).

CONCLUSION

  • clear, but give also the novelty of this study

In the conclusion section the novelty of our work was highlighted.

SUGGESTION

the MS is interesting with some significant results, but there are some imperfection that authors should correct before possible progress

OTHER COMMENTS

  • it is not clear if the MS is correctly related to aim of the journal (this should be decided by editor), but the MS is interesting describing some new ideas

The Special Issue of Materials is entitled ‘Preparation, Physico-chemical Properties and Biomedical Applications of Nanoparticles’. In our MS, we presented the preparation and physico-chemical characterization of SeNPs (80 nm) and we demonstrated the ability of its to diminish the toxic effects of Cd on liver tissue, which could be considered the biomedical application of these nanoparticles and fits with the aim of the special issue.

Reviewer 2 Report

You can find my comments below which are related to the reasons for my opinion.

What is the industrial use of the results obtained and perspectives for the future - please write a short commentary in the article.

Please describe the properties of selenium as a compound with anti-oxidation properties in introduction. The language of the manuscript is not adequate for the desired journal level. The text may be handled by a native English speaker.

Abstract is not clear for the reader and do not summarize the work. For instance: no introduction to the topic. It should be reformulated. Add more details, emphasize the importance of the results. In the abstract, there is no chemical form of selenium? The authors should average this throughout the work.

Selenium is a very interesting element, but also a dangerous one. It all depends on the offer used and the chemical form. The authors use the results of the opinion publication. The authors extend the introduction of selenium to chemical forms and its occurrence in various food products.

This needs to be seriously addressed as certain statements in their current context are thoroughly confusing. Discussion is speculative and based on the literature not on the own findings (The process of accumulation of selenium occurs in various ways to the cells. (conversion of selenium are poorly presented). It seems to be in some imbalance with Introduction. When discussing the results obtained in the previous studies the authors should identify which selenium species they are talking about

Author Response

Response to Reviewer 2

The authors would like to thanks for Reviewer 2 comments that improve our manuscript quality. 

(x) Extensive editing of English language and style required

The manuscript was checked for type-errors and English reviewed.

  • What is the industrial use of the results obtained and perspectives for the future - please write a short commentary in the article.

The possible industrial use was highlighted at the end of the MS.

  • Please describe the properties of selenium as a compound with anti-oxidation properties in introduction.

The antioxidant properties of selenium were described between 76-86 lines and we added new phrases in the subsections 3.3 and 3.4.

  • The language of the manuscript is not adequate for the desired journal level. The text may be handled by a native English speaker.

The manuscris was checked for type-errors and English reviewed.

  • Abstract is not clear for the reader and do not summarize the work. For instance: no introduction to the topic. It should be reformulated. Add more details, emphasize the importance of the results. In the abstract, there is no chemical form of selenium? The authors should average this throughout the work.

In the abstract we introduced an introductory phrase, we rephrased the results and we introduced a phrase about the importance of the results. Also, we added the chemical form of the selenium used.

The chemical form of SeNPs is elemental Se0. In order to produce SeNPs, we started from sodium hydrogen selenite (NaHSeO3), the protocol is described in Materials and methods. At the end of fermentation process, the appearance of red colour in medium indicates the elemental selenium producing. In the Introduction Chapter we introduced informations about different allotropic forms of Se. The XRD pattern of the Se nanoparticles (Figure 2c) shows a broad peak at about 2θ = 22° suggesting the amorphous structure of SeNPs.

  • Selenium is a very interesting element, but also a dangerous one. It all depends on the offer used and the chemical form. The authors use the results of the opinion publication. The authors extend the introduction of selenium to chemical forms and its occurrence in various food products.

In introduction chapter we introduced details regarding to the chemical forms (mainly organic forms) and its occurrence in food products.

  • This needs to be seriously addressed as certain statements in their current context are thoroughly confusing. Discussion is speculative and based on the literature not on the own findings (The process of accumulation of selenium occurs in various ways to the cells. (conversion of selenium are poorly presented). It seems to be in some imbalance with Introduction. When discussing the results obtained in the previous studies the authors should identify which selenium species they are talking about.

We changed the discussion according to your suggestion.

Reviewer 3 Report

Interesting study and concept with great potential for improvement of human health.  Needs further discussion, grammar edits and the possibly the addition of an experiment to address possible alterations in metallothionein levels.

Materials and Methods

  1. Line 97- 99: “Lactobacillus casei (Lyofast LC4P1, Sacco) was selected for SeNPs synthesis via reduction route using sodium hydrogen selenite (NaHSeO3) as a reducing agent, according to a protocol described by [21].”  This sentence ends abruptly – authors may want to consider ending the sentence as follows….”according to a protocol described by AUTHORS NAME [21].”
  2. Why were only female mice used?
  3. What kind of food were the mice fed? Normal chow tends to contain varying levels of heavy metals and therefore it is recommended that researchers use purified diets to ensure low levels of heavy metals in the diet.

Results and Discussion

  1. I suggest breaking apart Figure 1 into three separate panels and not have them overlapping in the same picture.
  2. I suggest that in table 2, under the group name in each column the authors indicate the details for each group (Cd and Se treatments) as it is difficult to remember all 8 groups when looking at these tables.
  3. Keep the formatting consistent – either maintain spaces between the measurements and units or take them out [i.e AST (U/L) vs ALT(U/L)]
  4. How many animals were considered in the H&E analysis per a group? Were there any outliers or where the results consistent across all animals in a group?
  5. Lines 343 and 344: The authors did not measure cadmium levels in the liver and therefore cannot make the statement…. “The present study demonstrated that co-treatment with SeNP ameliorated histopathological damage induced by Cd in the tissues of liver by reducing the toxicity and absorption of Cd.”
  6. The mention the importance of metallothionein (MT) but do not measure it therefore it seems a little our place. Measuring the MT levels in each group would strengthen there conclusions.
  7. There need so be more discussion and elaboration on the results and how it relayed to the existing literature. For example – why was the 5 mg/kg picked for cadmium? Why 30 days? Why via oral gavage? Also, how do these doses related to humans?

Author Response

Response to Reviewer 3

The authors would like to thanks for Reviewer 3 comments that improve our manuscript quality. 

 (x) Moderate English changes required

The manuscris was checked for type-errors and English reviewed .

Comments and Suggestions for Authors

  • Interesting study and concept with great potential for improvement of human health. Needs further discussion, grammar edits and the possibly the addition of an experiment to address possible alterations in metallothionein levels.

Materials and Methods

  1. Line 97- 99: “Lactobacillus casei (Lyofast LC4P1, Sacco) was selected for SeNPs synthesis via reduction route using sodium hydrogen selenite (NaHSeO3) as a reducing agent, according to a protocol described by [21].” This sentence ends abruptly – authors may want to consider ending the sentence as follows….”according to a protocol described by AUTHORS NAME [21].”

The problem was solved.

  1. Why were only female mice used?

To date, cadmium retention is generally higher in women than in men. Gender differences in susceptibility at lower exposure are uncertain, but recent data indicate that cadmium has estrogenic effects and affect female offspring (Vahter et al, 2007). Thus, we evaluated the hepatic protection of NpSe on females, which appear to be more susceptible to the toxic effects of cadmium.

Marie Vahter, Agneta Akesson, Carola Lidén, Sandra Ceccatelli, Marika Berglund, Gender differences in the disposition and toxicity of metals, Environ Res. 2007 May;104(1):85-95.

  1. What kind of food were the mice fed? Normal chow tends to contain varying levels of heavy metals and therefore it is recommended that researchers use purified diets to ensure low levels of heavy metals in the diet.

Mice were fed with an autoclavable standard scientific diet for rodents (Safe D40 diet, SAFE Complete Care Competence, Germany), which is certified free of toxic substances and balanced regarding the content of amino acids, fatty acids, minerals and vitamins.

Results and Discussion

  1. I suggest breaking apart Figure 1 into three separate panels and not have them overlapping in the same picture.

Figure 1 has been reconsidered and images are separated in 3 different panels.

  1. I suggest that in table 2, under the group name in each column the authors indicate the details for each group (Cd and Se treatments) as it is difficult to remember all 8 groups when looking at these tables.

In Table 2 , under the group name in each column we indicated the details for each groups. The details for each group were also specified in the Figures.

  1. Keep the formatting consistent – either maintain spaces between the measurements and units or take them out [i.e AST (U/L) vs ALT(U/L)]

We fixed the problem.

  1. How many animals were considered in the H&E analysis per a group? Were there any outliers or where the results consistent across all animals in a group?

We added to material and method section.

  1. Lines 343 and 344: The authors did not measure cadmium levels in the liver and therefore cannot make the statement…. “The present study demonstrated that co-treatment with SeNP ameliorated histopathological damage induced by Cd in the tissues of liver by reducing the toxicity and absorption of Cd.”

We changed with: “The present study demonstrated that co-treatment with SeNPs ameliorated histopathological damage induced by Cd in the tissues of liver by reducing the toxicity comparative with control group”

  1. The mention the importance of metallothionein (MT) but do not measure it therefore it seems a little our place. Measuring the MT levels in each group would strengthen there conclusions.

According to the literature date (Satarug, 2018), Cd can exert toxicity as a free ion. Cd intake from food is transported via hepatic portal system to the liver, where it induces the synthesis of MT. Cd becomes tightly bound to MT, and this complex is viewed as detoxified form. But the liver cells do not take up the complex, CdMT from the gastrointestinal tract may be transported directly to kidneys.

We consider that the measure of MT is a marker of Cd toxicity but the important target is kidneys. In the future, we will consider this analysis in our activities.

  1. There need so be more discussion and elaboration on the results and how it relayed to the existing literature. For example – why was the 5 mg/kg picked for cadmium? Why 30 days? Why via oral gavage? Also, how do these doses related to humans?

The dose of Cd used in this study (5 mg/kg b.w.) was choose based on the literature data that suggesting that the dose induces toxic effects. The three doses of SeNPs (0.1, 0.2, 0.4 mg/kg b.w.) and route of administration were selected according to the results in which protection was obtained against cadmium, administered to mice at a dose of 5 mg/kg (Ren et al, 2012).

Round 2

Reviewer 1 Report

As authors have done a serious revision and corrected the MS according to suggestions, the MS should be accepted for publication.

Author Response

Dear reviewer,

Your comments and suggestions have led to the improvement of our article.

Thank you for your effort in evaluating our article.

Respectfully,

Prof. dr. Simona Vicas

Reviewer 2 Report

The article has been corrected. I accept

Author Response

(The authors gave the same response as above.)

Reviewer 3 Report

1. Based on my previous question of “Why were only female mice used?” the author answered with the following…

“To date, cadmium retention is generally higher in women than in men. Gender differences in susceptibility at lower exposure are uncertain, but recent data indicate that cadmium has estrogenic effects and affect female offspring (Vahter et al, 2007). Thus, we evaluated the hepatic protection of NpSe on females, which appear to be more susceptible to the toxic effects of cadmium.”

I would like to make two points.

First, literature, both epidemiological and experimental, have shown males are more susceptible cadmium-induced hepatotoxicity (Hyder et al 2013, Lonardo et al., 2019). For example, Hyder et al 2013 showed cadmium exposure was associated with hepatic necroinflammation, NAFLD, and NASH in men, and only hepatic necroinflammation in women.

Hyder, O., Chung, M., Cosgrove, D., Herman, J. M., Li, Z., Firoozmand, A., Gurakar, A., Koteish, A., & Pawlik, T. M. (2013). Cadmium exposure and liver disease among US adults. Journal of gastrointestinal surgery : official journal of the Society for Surgery of the Alimentary Tract, 17(7), 1265–1273. https://doi.org/10.1007/s11605-013-2210-9.

Lonardo, A., Nascimbeni, F., Ballestri, S., Fairweather, D., Win, S., Than, T. A., Abdelmalek, M. F., & Suzuki, A. (2019). Sex Differences in Nonalcoholic Fatty Liver Disease: State of the Art and Identification of Research Gaps. Hepatology (Baltimore, Md.), 70(4), 1457–1469. https://doi.org/10.1002/hep.30626

Second, I recommend putting a sentence in the methods as to why you choose only female mice. I do not thin for this paper to be published that male mice have to be included, but I do think an explanation for the readers is warranted and possibly a statement in the end of the discussion about follow-up studies with males and what different outcomes you may find.

2. Based on my previous question, “What kind of food were the mice fed? Normal chow tends to contain varying levels of heavy metals and therefore it is recommended that researchers use purified diets to ensure low levels of heavy metals in the diet?” the authors replied…..

“Mice were fed with an autoclavable standard scientific diet for rodents (Safe D40 diet, SAFE Complete Care Competence, Germany), which is certified free of toxic substances and balanced regarding the content of amino acids, fatty acids, minerals and vitamins.”

I recommend adding the information on the diet in the methods section, at least state the name of the diet and as supplemental material included the certificate and/or diet components. When working with metals it is key to be very transparent with the diet and its components.

3. Double check that the spaces and formatting are consistent throughout the manuscript. For example in some parts of the manuscript there is space between the last word of a sentence and the references and in other parts there is no space. 

Author Response

Dear Reviewer,

  1. Based on my previous question of “Why were only female mice used?” the author answered with the following…

“To date, cadmium retention is generally higher in women than in men. Gender differences in susceptibility at lower exposure are uncertain, but recent data indicate that cadmium has estrogenic effects and affect female offspring (Vahter et al, 2007). Thus, we evaluated the hepatic protection of NpSe on females, which appear to be more susceptible to the toxic effects of cadmium.” 

I would like to make two points.

First, literature, both epidemiological and experimental, have shown males are more susceptible cadmium-induced hepatotoxicity (Hyder et al 2013, Lonardo et al., 2019). For example, Hyder et al 2013 showed cadmium exposure was associated with hepatic necroinflammation, NAFLD, and NASH in men, and only hepatic necroinflammation in women.

Hyder, O., Chung, M., Cosgrove, D., Herman, J. M., Li, Z., Firoozmand, A., Gurakar, A., Koteish, A., & Pawlik, T. M. (2013). Cadmium exposure and liver disease among US adults. Journal of gastrointestinal surgery : official journal of the Society for Surgery of the Alimentary Tract, 17(7), 1265–1273. https://doi.org/10.1007/s11605-013-2210-9.

Lonardo, A., Nascimbeni, F., Ballestri, S., Fairweather, D., Win, S., Than, T. A., Abdelmalek, M. F., & Suzuki, A. (2019). Sex Differences in Nonalcoholic Fatty Liver Disease: State of the Art and Identification of Research Gaps. Hepatology (Baltimore, Md.), 70(4), 1457–1469. https://doi.org/10.1002/hep.30626

Second, I recommend putting a sentence in the methods as to why you choose only female mice. I do not thin for this paper to be published that male mice have to be included, but I do think an explanation for the readers is warranted and possibly a statement in the end of the discussion about follow-up studies with males and what different outcomes you may find.

 Response: We agree with you and thank you for your suggestions. We introduced a sentence in the methods with justification of choosing female in the study and a sentence to the end in which we mention that follow-up studies on males should be addressed to highlight the gender differences in the hepatoprotective effects of SeNPs, knowing that males are more susceptible to cadmium-induced hepatotoxicity than female (Hyder et al 2013, Lonardo et al., 2019).

Hyder, O., Chung, M., Cosgrove, D., Herman, J. M., Li, Z., Firoozmand, A., Gurakar, A., Koteish, A., & Pawlik, T. M. (2013). Cadmium exposure and liver disease among US adults. Journal of gastrointestinal surgery : official journal of the Society for Surgery of the Alimentary Tract, 17(7), 1265–1273. https://doi.org/10.1007/s11605-013-2210-9.

Lonardo, A., Nascimbeni, F., Ballestri, S., Fairweather, D., Win, S., Than, T. A., Abdelmalek, M. F., & Suzuki, A. (2019). Sex Differences in Nonalcoholic Fatty Liver Disease: State of the Art and Identification of Research Gaps. Hepatology (Baltimore, Md.), 70(4), 1457–1469. https://doi.org/10.1002/hep.30626

  1. Based on my previous question, “What kind of food were the mice fed? Normal chow tends to contain varying levels of heavy metals and therefore it is recommended that researchers use purified diets to ensure low levels of heavy metals in the diet?” the authors replied…..

“Mice were fed with an autoclavable standard scientific diet for rodents (Safe D40 diet, SAFE Complete Care Competence, Germany), which is certified free of toxic substances and balanced regarding the content of amino acids, fatty acids, minerals and vitamins.”

I recommend adding the information on the diet in the methods section, at least state the name of the diet and as supplemental material included the certificate and/or diet components. When working with metals it is key to be very transparent with the diet and its components.

 Response: We added to name of the died and the provided company.

  1. Double check that the spaces and formatting are consistent throughout the manuscript. For example in some parts of the manuscript there is space between the last word of a sentence and the references and in other parts there is no space.

Response: We double check that the spaces and formatting are consistent throughout the manuscript.
